# Hypermethylation of the Gene Body in *SRCIN1* Is Involved in Breast Cancer Cell Proliferation and Is a Potential Blood-Based Biomarker for Early Detection and a Poor Prognosis

**DOI:** 10.3390/biom14050571

**Published:** 2024-05-12

**Authors:** Hsieh-Tsung Shen, Chin-Sheng Hung, Clilia Davis, Chih-Ming Su, Li-Min Liao, Hsiu-Ming Shih, Kuan-Der Lee, Muhamad Ansar, Ruo-Kai Lin

**Affiliations:** 1The Ph.D. Program for Translational Medicine, College of Medical Science and Technology, Taipei Medical University and Academia Sinica, Taipei 110301, Taiwan; ethanshen@eg-bio.com (H.-T.S.); hmshih@ibms.sinica.edu.tw (H.-M.S.); kdlee@tmu.edu.tw (K.-D.L.); 2EG BioMed US Inc., Covina, CA 91722, USA; hungcs@tmu.edu.tw; 3Department of Surgery, School of Medicine, College of Medicine, Taipei Medical University, Taipei 110301, Taiwan; 08261@shh.org.tw; 4Division of General Surgery, Department of Surgery, Shuang Ho Hospital, Taipei Medical University, New Taipei City 235041, Taiwan; 18643@s.tmu.edu.tw; 5Division of Breast Surgery, Department of Surgery, Taipei Medical University Hospital, Taipei 110301, Taiwan; 6International Master Program in Medicine, College of Medicine, Taipei Medical University, Taipei 110301, Taiwan; m142110004@tmu.edu.tw; 7Comprehensive Cancer Center, Taichung Veterans General Hospital, Taichung 40705, Taiwan; 8Ph.D. Program in the Clinical Drug Development of Herbal Medicine, Taipei Medical University, Taipei 110301, Taiwan; 9Graduate Institute of Pharmacognosy, Ph.D. Program in Drug Discovery and Development Industry, Masters Program for Clinical Pharmacogenomics and Pharmacoproteomics, College of Pharmacy, Taipei Medical University, Taipei 110301, Taiwan; 10Clinical Trial Center, Taipei Medical University Hospital, Taipei 110301, Taiwan

**Keywords:** *SRCIN1*, breast cancer, DNA methylation, circulating cell-free DNA, RNA-seq, biomarker

## Abstract

Breast cancer is a leading cause of cancer mortality in women worldwide. Using the Infinium MethylationEPIC BeadChip, we analyzed plasma sample methylation to identify the *SRCIN1* gene in breast cancer patients. We assessed *SRCIN1*-related roles and pathways for their biomarker potential. To verify the methylation status, quantitative methylation-specific PCR (qMSP) was performed on genomic DNA and circulating cell-free DNA samples, and mRNA expression analysis was performed using RT‒qPCR. The results were validated in a Western population; for this analysis, the samples included plasma samples from breast cancer patients from the USA and from The Cancer Genome Atlas (TCGA) cohort. To study the *SRCIN1* pathway, we conducted cell viability assays, gene manipulation and RNA sequencing. *SRCIN1* hypermethylation was identified in 61.8% of breast cancer tissues from Taiwanese patients, exhibiting specificity to this malignancy. Furthermore, its presence correlated significantly with unfavorable 5-year overall survival outcomes. The levels of methylated *SRCIN1* in the blood of patients from Taiwan and the USA correlated with the stage of breast cancer. The proportion of patients with high methylation levels increased from 0% in healthy individuals to 63.6% in Stage 0, 80% in Stage I and 82.6% in Stage II, with a sensitivity of 78.5%, an accuracy of 90.3% and a specificity of 100%. *SRCIN1* hypermethylation was significantly correlated with increased *SRCIN1* mRNA expression (*p* < 0.001). Knockdown of SRCIN1 decreased the viability of breast cancer cells. *SRCIN1* silencing resulted in the downregulation of ESR1, BCL2 and various cyclin protein expressions. *SRCIN1* hypermethylation in the blood may serve as a noninvasive biomarker, facilitating early detection and prognosis evaluation, and *SRCIN1*-targeted therapies could be used in combination regimens for breast cancer patients.

## 1. Introduction

Breast cancer (BC) remains one of the most common cancers globally and is the leading cause of cancer-related death in women [1]. Breast cancer is indeed a complex disease. Its complexity is influenced by various factors, such as the cancer type, risk factors, genetics, hormonal influences, personalized treatments and lifestyle factors [2,3,4,5,6,7,8,9,10]. The presentation of these factors can vary among individuals, contributing to the complexity of both diagnosis and treatment. Therefore, early detection, routine screenings and persistent research is essential to expand our understanding of this disease and improve treatment outcomes.

The mortality rate has decreased by 20% due to the use of imaging-based techniques such as mammography [11]. Nevertheless, these methods have limitations, including reduced sensitivity, high false-positive rates and patient discomfort [12]. In addition, the invasiveness and sampling biases associated with tissue biopsies pose challenges to effective cancer detection and monitoring [13,14]. Currently, the prognosis of this disease is determined using imaging and immunohistochemical analyses [15]. However, these markers are not always definitive or highly sensitive indicators of disease outcomes [16]. Successful breast cancer management and the prevention of adverse outcomes rely on accurate and early disease diagnosis. Currently, there are no established independent noninvasive molecular epigenetic markers for breast cancer diagnosis and treatment [17]. Epigenetic profiling of blood plasma via liquid biopsies, particularly through DNA methylation analysis, is emerging as a pivotal method for breast cancer detection and monitoring due to its stability and distinct modification patterns, offering promising avenues for diagnostic and prognostic biomarker discovery [18]. For example, global hypomethylation and hypermethylation of CpG sites occur early in BC, highlighting possible markers of early-stage disease with increased sensitivity [19]. Furthermore, in various studies, scholars have outlined the feasibility of gene-specific DNA methylation alterations as prognostic biomarkers [20,21]. Several biomarkers have been used for breast cancer diagnosis, prognosis evaluation and treatment decision making. Common factors include hormone receptor and HER2 status, Ki-67 staining, *BRCA1/BRCA2* mutations and MammaPrint results [22,23]. Additionally, targeting DNA methylation on specific genes holds great promise as an epigenetic therapy for breast cancer [24]. Therefore, the development of candidate biomarkers to guide treatment decision making is a key objective of this study.

In response to the substantial unmet medical need, this study was designed to explore the potential of plasma-based biomarkers for early diagnosis and prognosis evaluation and for the identification of the potential of drug target genes in breast cancer. For our investigation, a human Infinium methylation EPIC array containing plasma samples from Taiwanese breast cancer patients was used to identify a novel hypermethylated gene, SRC kinase signaling inhibitor 1 (*SRCIN1/p140cap*), specific to breast cancer. In this study, we sought to elucidate the methylation levels, expression patterns and biological functions of *SRCIN1*. Additionally, we assessed the high-specificity detection of circulating methylated *SRCIN1* in the blood of Taiwanese and Western patients with breast cancer.

## 2. Materials and Methods

### 2.1. Patient Breast Tissue and Plasma Collection

We obtained a total of 12 plasma samples from healthy subjects, 24 plasma samples from breast cancer patients at stage 0 to stage I and 5 paired tumor and normal tissue samples were used for genome-wide methylation analysis, 102 paired tissue samples for *SRCIN1* gene-specific methylation assays and 43 matched pairs were procured for *SRCIN1* mRNA expression assays. All sample were collected from Taipei Medical University (TMU) Hospital, Shuang Ho Hospital, and the TMU-Joint Biobank for DNA methylation analysis. Patients were enrolled for sample collection after they read, understood and signed the informed consent form, indicating that they agreed to participate in the study. The patients’ clinical data, including race, tumor classifications, personal and family medical history, tumor location, TNM stage and follow-up conditions, were prospectively obtained. Various sections of diseased and adjacent normal tissues were reviewed by a senior pathologist.

In the context of the gene-specific cell-free DNA methylation assay, a prospective study was conducted in Taipei, Taiwan; we enrolled a total of 4 healthy subjects and 23 patients between 2016 and 2023. These patients were carefully monitored and included individuals with breast cancer with stage 0 to stage II tumors. The study participants were patients of the Breast Medical Center at Taipei Medical University Hospital as well or the Breast Medical Center and Division of Hematology/Oncology at Shuang Ho Hospital. Written informed consent was obtained from all patients prior to the collection of clinical data and samples. The study samples included 5 plasma samples from healthy controls and 61 plasma samples obtained from breast cancer patients in the United States. These samples were collected by Precision for Medicine, LLC. (Norton, MA, USA).

### 2.2. Genomic DNA, Circulating Cell-Free DNA and RNA Extraction

Matched pairs of tumor and normal tissues were collected from each patient following surgery, immediately stored at −80 °C and then transferred to liquid nitrogen. Genomic DNA from 102 tissue pairs and messenger RNA from 43 tissue pairs were extracted using the QIAamp DNA Mini Kit (Qiagen, Bonn, Germany, Cat. No. 51306) and the RNeasy Plus Mini Kit (Qiagen, Bonn, Germany, Cat. No. 74134), respectively. The RNA and DNA were quantified, and the purity was verified by measuring the A260/A280 ratio (which ranged from 1.8 to 2.0) [25] using a Thermo Scientific™ NanoDrop 2000c (Thermo Fisher Scientific, Waltham, MA, USA.) For further experimentation, genomic DNA was then stored at −20 °C, while mRNA was stored at −80 °C.

Blood samples were collected using Streck BCT or PAXgene cfDNA tubes, and a double centrifugation process was used to isolate plasma. Circulating cell-free DNA (cfDNA) was extracted from 63 of the plasma samples using the iCatcher Circulating cfDNA 1000 kit (CatchGene, New Taipei City, Taiwan) according to the manufacturer’s recommended protocol.

### 2.3. DNA Methylation Array Assay

Genome-wide methylation analysis of plasma was conducted using the Infinium MethylationEPIC 850K array Kit (Illumina). To ensure an adequate amount and quality of cfDNA, the samples were prepared by pooling plasma from 12 healthy Taiwanese individuals in one reaction, 12 Taiwanese stage I breast cancer patients in another reaction and 12 Taiwanese stage 0 breast cancer patients in a third reaction, followed by methylation array analysis.

Genome-wide methylation analysis of 5 paired tumor and normal breast cancer tissues was performed using the Illumina Infinium HumanMethylation450 BeadChip array (Illumina, San Diego, CA, USA) for one sample and the Infinium MethylationEPIC Kit (Illumina) for the remaining 4 samples, as previously reported [26,27]. The two arrays contain more than 450,000 and 850,000 methylation sites, respectively, and provide genome-wide coverage of the gene region and CpG island coverage, respectively, including 99% of RefSeq genes.

Methylation levels from plasma or tissue array were quantified as beta values, which reflect the ratio of methylated signal intensity to the total signal intensity, encompassing both methylated and unmethylated signals. These values ranged from 0, indicating no methylation, to 1, indicating complete methylation.

### 2.4. Gene-Specific DNA Methylation Assay Using Quantitative Methylation-Specific PCR (qMSP)

The 500 ng of genomic DNA and 75–150 ng of cfDNA were subjected to bisulfite treatment using the EpiTect Fast DNA Bisulfite Kit (Qiagen, Bonn, Germany, Catalog No. 59826), after which the methylation level of the *SRCIN1* gene was quantified. This quantification was performed using qMSP techniques on a LightCycler 96 system (Roche Applied Science, Penzberg, Germany). For qMSP, the SensiFAST™ Probe No-ROX Kit (Bioline, London, UK, Catalog No. BIO86005) and primers and probes specific for *SRCIN1* were used. Methylation values were standardized against those of an internal control and interpreted using LightCycler 96 Relative Quantification software (version 2.0, Roche Applied Science). All experimental assays were performed in triplicate to ensure statistical robustness.

In genomic DNA methylation analysis from tissues, *SRCIN1* was considered hypermethylated when the methylation level of *SRCIN1* relative to the reference gene beta-actin (ACTB) was at least two-fold higher in breast tumors than in normal breast tissue samples [28]. Moreover, in cfDNA methylation analysis from plasma, a relative methylation level of *SRCIN1* to *ACTB* greater than 0.04 indicated positive *SRCIN1* methylation. Samples not exceeding the threshold for a positive result were classified as ‘negative’. *SRCIN1* quantitative methylation-specific PCR primers and probes were designed using MethPrimer software 1.0 [29]. The primers and probes used for qMSP are listed in Appendix A.

### 2.5. Gene Expression Assays Using Reverse Transcriptase qPCR

Reverse transcriptase PCR was performed with FIREScript RT cDNA synthesis mix (Solis Byodine, Tartu, Estonia, Cat. No 06-20-00100). A total of 2 μg of purified mRNA was mixed with the correct nucleotides and enzymes. The solution was then placed in a Labcycler 48 thermal cycler (Siemens, Gottingen, Germany) at appropriate temperatures according to the manufacturer’s recommended protocol. Complementary DNA (cDNA) was then stored at −20 °C for further experimentation.

The product for the reverse transcriptase reaction cDNA was used to determine the mRNA expression level via reverse transcriptase qPCR using a LightCycler 96 (Roche Applied Science, Mannheim, Germany). In this study, specific primers and probes at appropriate concentrations were added to template cDNA. Gene expression was normalized to that of the internal control glyceraldehyde 3-phosphate dehydrogenase (*GAPDH*) using LightCycler Relative Quantification software (ver. 2.0, Roche Applied Science). *SRCIN1* mRNA expression was considered high if the relative expression level (normalized to that of *GAPDH*) was more than 2 times greater in tumors than in normal breast tissue. However, target genes were considered to have low expression if the relative expression level of the target gene (normalized to that of *GAPDH*) was less than 0.5 times lower in tumors than in normal breast tissue [27]. The primers and probes used for RT‒qPCR are listed in Appendix A.

### 2.6. Cell Culture, Counting and RNAi Transfection

MCF-7, T47D and MDA-MB-231 breast cancer cell lines utilized in this study were acquired from the Bioresource Collection and Research Center (BCRC, Hsinchu, Taiwan). MCF-7, T47D and MDA-MB-231 cells were cultured in DMEM/F12 supplemented with 5%, 7.5% and 2.5% human platelet lysate, respectively, and 1% penicillin/streptomycin at 37 °C and 5% CO_2_. After the cells were transfected, RNA was extracted for cell viability and gene expression analysis. After the MCF-7, T47D and MDA-MB-231 breast cancer cell lines were seeded, they were subsequently counted with a microscope and hemocytometer. The experiments were completed in triplicate.

MCF-7, T47D and MDA-MB-231 cells were treated with 20 μM small interfering RNA (si-*SRCIN1*; Ambion, Austin, TX, USA; Cat No.: 4392420). Simultaneously, cells were treated with 20 μM negative control #1 (si-control #1, Ambion, USA, Cat No.: 4390843) as the control group. All siRNA transfection reactions were completed with Lipofectamine 3000 Reagent (Invitrogen, Waltham, MA, USA, Cat No.: 100022052) and P3000 Reagent (Invitrogen, MA, USA, Cat No.: 100022058).

To assess *SRCIN1* knockdown efficiency, RNA was extracted from treated cell lysates using TRIzol reagent, and a total of 2 μg of purified mRNA was reverse transcribed into cDNA (Solis Biodyne, FIREScript^®^ RT cDNA Lot No. 0620001807). RT‒qPCR was conducted as described in Section 2.5 above.

For the *SRCIN1* demethylation assay, T47D and MCF-7 cells were treated with dimethyl sulfoxide (DMSO) or with 10 μM demethylation agent decitabine (DAC, Sigma–Aldrich, St. Louis, MO, USA) for 96 h. DAC was dissolved in DMSO. After treatment of the cells, DNA and RNA were extracted, and methylation and gene expression levels were analyzed.

### 2.7. Cell Viability Assay

Cell viability was assessed with the MTT (3-(4,5-dimethylthiazol-2-yl)-2,5-diphenyltetrazolium bromide) assay. A total of 8 × 10^3^ cells were seeded in each well of a 96-well plate and then transfected with si-*SRCIN1* for 48 h. After transfection, the cells were treated with 0.5 mg/mL MTT, and the cells were incubated for 4 h at 37 °C. The absorbance of each well was measured at 570 nm utilizing a Thermo Scientific Varioskan Flash Multimode Microplate Reader. Untreated cells served as controls. The experiments were repeated to obtain at least three biological replicates. Pictures were taken at 4× magnification.

### 2.8. RNA Sequencing and Signaling Pathway Analysis

Total mRNA was extracted from MCF-7 breast cancer cells transfected with 20 μM small interfering RNA or 20 μM negative control #1 as the control group. The purified RNA was utilized for the preparation of the sequencing library with the TruSeq Stranded mRNA Library Prep Kit (Illumina, San Diego, CA, USA) following the manufacturer’s instructions. Then, mRNA was purified from total RNA (1 μg) by oligo(dT)-coupled magnetic beads and broken into smaller segments with increasing temperature. First-strand cDNA was generated with reverse transcriptase and random primers. After the synthesis of double-stranded cDNA and adenylation of the 3′ ends of the DNA fragments, the adaptors were ligated and purified with an AMPure XP system (Beckman Coulter, Beverly, CA, USA). Quality assurance of the libraries was completed using an Agilent Bioanalyzer 2100 system and a real-time qPCR system. Then, the qualified libraries were sequenced on an Illumina NovaSeq 6000 platform with 150 bp paired-end reads generated by Genomics, BioSci & Tech Co., New Taipei City, Taiwan. Next, we calculated the log2 ratio as the difference in the log2 expression of si-*SRCIN1* and si-control. *SRCIN1*-related pathways and networks in cancer were analyzed via the MetaCore repository (Clarivate Analytics, Philadelphia, PA, USA) [30].

### 2.9. The Cancer Genome Atlas Portal and Data Analysis

The Cancer Genome Atlas (TCGA) Research Network (http://cancergenome.nih.gov/ accessed on 6 June 2016) was used to retrieve DNA methylation, expression status, and clinical data for the Western cohort. The TCGA has been a powerful tool for the preliminary study and identification of important cancer-related alterations, with coverage of DNA, RNA, miRNA and protein levels in more than 83,000 cancer cases of 33 cancer types, among which 9118 cases are breast cancer cases (https://portal.gdc.cancer.gov/) which were downloaded between 2015 and 2021. For this study, the DNA methylation level was considered to be low when the methylation level of the tumor tissue was equal to or less than half of the average methylation level of the adjacent normal tissue. Additionally, DNA methylation was considered high when the mean methylation level of the tumor tissue was double or more than double that of the mean methylation level of the adjacent normal tissue. mRNA expression was considered low when the expression level in the tumor tissue was half or less than half of that in the adjacent normal tissue and high when the expression level in the tumor tissue was double or more than double that in the adjacent normal tissue. We utilized the heatmapper program (http://www.heatmapper.ca/expression/ accessed on 28 April 2024) to create the heatmaps, the heatmaps have been customized to depict high, middle and low DNA methylation levels, utilizing a gradient that transitions from blue to yellow [31,32,33,34].

The analysis of the Western cohort was The Cancer Genome Atlas (TCGA) Research Network data from the Genomic Data Commons (GDC) data portal (https://portal.gdc.cancer.gov/), which were downloaded between 2015 and 2021. For DNA methylation analysis, Illumina Infinium HumanMethylation450 BeadChip array data from 640 breast tumors, 87 paired tissues from breast cancer patients were analyzed. For RNA expression analysis, RNA-seq data were obtained for 755 breast tumors and 57 paired tissues from breast cancer patients.

### 2.10. Statistical Analysis

SPSS (SPSS Inc., Chicago, IL, USA) was utilized for all the statistical analyses. The Pearson *X*^2^ test was used to evaluate all clinical data in concordance with the *SRCIN1* DNA methylation and mRNA expression profiles. The clinical parameters included age, race, menopausal stage, histological type, tumor stage, tumor size, lymph node status, metastasis status and molecular subtype. The correlation between DNA methylation and the mRNA expression of *SRCIN1* was determined via Spearman correlation analysis. Additionally, to assess differences in DNA methylation and mRNA expression between paired breast tissue samples and cell viability, paired *t* tests were performed. In addition, the significance of survival associations was computed using the log-rank test. In addition to accuracy, other commonly used measures for evaluating classification, such as area under the receiver operating characteristic (ROC) curve (AUC), sensitivity, specificity, false-positive rate and false-negative rate, were also reported.

## 3. Results

### 3.1. SRCIN1 Was Identified through Genome-Wide Methylation and RNA Expression Analysis

To identify novel potential biomarkers for early prediction and drug targets in breast cancer patients from both Taiwanese and Western cohorts, we have followed the present strategy to screen for targets: (1) genome-wide methylation profiling of pooled plasma from healthy Taiwanese individuals (*n* = 12), pooled plasma from Taiwanese Stage I breast cancer patients (*n* = 12) and pooled plasma from Taiwanese stage 0 breast cancer patients (*n* = 12), using the Infinium MethylationEPIC array; (2) analysis of methylation levels based on TCGA data for 87 paired samples from Western breast cancer patients; and (3) examination of TCGA RNAseq data for 72 paired tissue samples from breast cancer patients (Figure 1A and Appendix A).

First, in our efforts to pinpoint distinct markers in plasma, we established specific criteria based on β values. (1) A total of 6997 genes exhibited Avg_β values below 0.05 (low methylation level in healthy subjects). (2) The top 3000 genes with Δβ values for cfDNA methylation greater than 0.3 (relatively high methylation level) in pooled plasma samples from Stage I breast cancer patients vs. pooled samples from healthy individuals were selected for further analysis. Additionally, (3) the top 3000 genes with Δβ values for cfDNA methylation exceeding 0.3 (relatively high methylation level) in pooled plasma samples from breast cancer patients at stage 0 vs. samples from healthy patients were selected for further analysis.

Next, we analyzed the TCGA Illumina Infinium HumanMethylation450 BeadChip array data, focusing on 87 paired Western breast cancer patient tissues. Through our examination, we identified 1522 genes exhibiting hypermethylation based on the criterion of ΔAvg_β (β tumor − β normal)  >  0.3. Then, to identify potential oncogenes for drug targeting, we selected a minimal threshold of a 1.3-fold change to distinguish genes that exhibit overexpression at the RNA level compared to normal breast tissue expression profiles [35,36]; TCGA RNAseq data for 72 paired tissues were analyzed, and the results revealed 3227 upregulated genes according to the specified criterion (tumor/normal > 1.3). Using the InteractiVenn tool (Figure 1B), we identified 8 genes that met our criteria: *KIF5C*, *SFRP2*, *MSC*, *SHB*, *BCOR*, *CREB3L1*, *KIF26B* and *SRCIN1*. Methylation of *KIF5C* (kinesin family member 5C) is a promising biomarker for ulcerative colitis ulcerative-associated neoplasia [37], and methylation of the secreted frizzled-related protein 2 gene (*SFRP2*) is a promising noninvasive biomarker for diagnosing colorectal cancer through fecal sample analysis [38]. Musculin (*MSC*) DNA methylation plays a role in gastric carcinogenesis and holds potential as a valuable biomarker for diagnosing gastric cancer (GC) and detecting recurrence [39]. The *SHB* (SH2 domain containing adaptor protein B) region has been identified as hypermethylated in Graves’ disease [40]. Differential profiling of DNA methylation and copy number variations in pediatric tumors exhibiting *BCOR* (BCL6-correpresor) overexpression potentially contributes to diagnostic confirmation and stratification [41]. High-grade metastatic breast cancers with a poor prognosis exhibit epigenetic silencing of *CREB3L1* (cAMP-responsive element-binding protein 3-like protein 1) through DNA methylation, a phenomenon prevalent particularly in triple-negative breast cancers [42], and member 26B of the kinesin family (*KIF26B*) is widely recognized as an oncogene in breast, gastric, colorectal and hepatocellular cancers. Its overexpression in these malignancies significantly correlates with larger tumor sizes, heightened metastatic tendencies and a poor prognosis [43]. *SRCIN1* methylation has not been investigated in any disease, particularly in breast cancer, and its impact on the expression levels under these conditions remains unclear.

### 3.2. Hypermethylation of the SRCIN1 Gene Body and Elevated mRNA Expression in Taiwanese and TCGA Cohorts

To validate the observations of circulating methylated *SRCIN1* in breast cancer patients, as identified by a genome-wide methylation array in plasma, we conducted a methylation array analysis on five paired breast cancer tissue samples. This assay enabled the quantitative assessment of methylation status across numerous CpG sites in the tissue samples. There were differentially methylated CpG sites in the gene body, with increased methylation in the tumor cg14482902, cg01514828, cg03026373, cg14862207 and cg04194674 (Figure 2A). This observation was consistent with findings from the TCGA dataset, where a similar increase in methylation was noted in the tumor tissue at the gene body CpG sites (Figure 2B).

To further explore the significant hypermethylation patterns exhibited in individual breast cancer patients, we analyzed 102 paired tissue samples from Taiwanese breast cancer patients using quantitative methylation-specific PCR (qMSP). Significant hypermethylation was detected in 61.8% (63/102) of the samples (Figure 2C; Table 1; and Appendix A), and high expression of *SRCIN1* was detected in 48.8% of the patients (*p <* 0.05) (Figure 2D; and Table 1).

Further analysis of TCGA paired data revealed that cg14862207 was hypermethylated in 64.3% of tumor tissues compared to their corresponding normal tissues (Figure 2E; *p <* 0.001). The TCGA RNA-seq data revealed an increase in the expression of *SRCIN1* in 72 paired tumor tissue samples compared with normal tissue samples from breast cancer patients (Figure 2F; *p <* 0.001).

To investigate whether high expression levels of *SRCIN1* were modulated by DNA hypermethylation of *SRCIN1*, we treated breast cancer cells with the DNMT inhibitor 5-aza-2′-deoxycytidine (DAC) to suppress the methylation of the *SRCIN1* gene and then analyzed the *SRCIN1* mRNA expression level. Our data analysis revealed a decrease in methylation levels and a simultaneous reduction in expression levels in T47D cells (Figure 2G) and MCF7 cells (Figure 2H). These results substantiate the association between *SRCIN1* methylation and its expression.

We validated the relationship between methylation and expression levels in clinical data using the TCGA dataset. The relationships between the methylation of all the *SRCIN1* CpG sites and mRNA expression were analyzed using Spearman’s correlation test, and methylation at almost all the CpG sites was significantly correlated with mRNA expression. Strikingly, there was a positive and significant correlation between the hypermethylation of gene body CpG sites cg14482902, cg01514828, cg03026373, cg14862207 and cg04194674 and mRNA expression in the tumor tissue (Figure 2I and Appendix A). The opposite was observed for the selected differentially methylated promoter CpG sites cg06239352, cg02956148 and cg01156834, where hypomethylation in the tumor was inversely correlated with mRNA expression (Figure 2J; Appendix A).

### 3.3. Differential Methylation of the SRCIN1 Gene Is Specific to Breast Cancer and Are Correlated with Poor Survival

To assess the specificity of our marker as a breast cancer biomarker, we conducted methylation analysis across various cancer types in both the Taiwanese and TCGA datasets. We discovered that the methylation levels of *SRCIN1* in lung cancer, colon cancer, esophageal cancer and uterine cancer samples from Taiwanese patients were not significantly different between tumor and normal tissues (Figure 3A). Consistent with the Taiwanese data, the TCGA dataset also revealed that methylation at the five specific CpG sites of *SRCIN1* did not significantly differ among lung cancer, colon cancer, esophageal cancer, uterine cancer, liver cancer, pancreatic cancer and gastric cancer patients (Figure 3B and Appendix A).

Analysis of five-year survival in TCGA patients with *SRCIN1* methylation of all selected differentially methylated CpG sites in unpaired tumor breast tissue revealed that high methylation was associated with shorter survival than low methylation (Figure 3C; *p* < 0.024).

### 3.4. Detection of Circulating Methylated SRCIN1 in Breast Cancer Patients from Taiwan and the USA

To ensure the high specificity of the blood biomarker for detecting breast cancer, it is critical to have a low cfDNA methylation background in healthy subjects. Our investigations revealed that the methylation level of *SRCIN1* in cfDNA from plasma, when normalized with *ACTB* used as the cfDNA control, is consistently below 0.04 across all samples from healthy individuals (Figure 4A). Consequently, we have identified 0.04 as the threshold to differentiate between positive and negative tests.

The levels of methylated circulating *SRCIN1* showed a positive correlation with disease stage. The percentages of patients in the following groups who were positive for circulating methylated *SRCIN1* were as follows: healthy (0%), stage 0 (63.6%), stage I (80%) and stage II (82.6%). Overall, the marker had a sensitivity of 78.5%, an accuracy of 90.3%, a specificity of 100% and an AUC of 88.2%. (Figure 4A,B; and Table 2).

### 3.5. SRCIN1 Expression Was Significantly Correlated with ER, PR, HER2 and TNBC Status and Cell Viability

The percentage of patients with high expression of *SRCIN1* was higher in the ER-positive breast cancer (53.4%, *p* < 0.01) and PR-positive breast cancer (53.6%, *p* <0.05) groups than in the ER-negative and PR-negative groups. Among the 56 HER2-positive patients, a significant majority (66.1%, *p* < 0.01) had high *SRCIN1* expression. In contrast, although the majority of patients were not TNBC patients, among the TNBC patients, a substantial portion (43.3%, *p* < 0.001) had low expression of *SRCIN1* (Figure 5A; Appendix A).

These results are consistent with the cell data. RT‒qPCR revealed that in MCF7 cells after *SRCIN1* knockdown, the expression level of estrogen receptor 1 (*ESR1)* was decreased 38.7-fold (*p* < 0.001), while that of estrogen receptor 2 (*ESR2)* was decreased 1.8-fold (Figure 5B).

Silencing *SRCIN1* in MCF-7 cells led to a significant 68% decrease in cell viability (*p* < 0.001; Figure 5C). *SRCIN1* silencing led to a significant 14% decrease in cell viability (*p* < 0.05; Figure 5D) of T47D cells. Additionally, in MDA-MB231 cells, the silencing of *SRCIN1* led to a significant 33% decrease in cell viability (Figure 5E; *p* < 0.05).

### 3.6. SRCIN1 Silencing in the MCF-7 Cell Line Resulted in the Downregulation of the Estrogen Receptor, BCL2 and Cell Cyclin-Related Protein Pathways

To investigate the pathways regulated by *SRCIN1* expression in breast cancer, we performed gene silencing in the MCF-7 cell line and utilized RNA-seq analysis combined with MetaCore pathway analyses for data interpretation. Our RNA-seq results revealed a significant downregulation of two key genes, *ESR1* and *Bcl-2* (Figure 6A). Additionally, nine members of the cyclin protein family, Cyclin A, Cyclin A2, CKS, CDK6, CRM1, P27KIP1, SKP2, NCOA3 and CDK2, were also downregulated according to both RNA-seq (Figure 6B) and RT‒qPCR (Figure 6C) after transfection of MCF7 cells with si-*SRCIN1*.

## 4. Discussion

Cell-free DNA exhibiting irregular methylation patterns in body fluids could be considered a potential diagnostic and prognostic tool for managing breast cancer [44]. Nonetheless, while the FDA has granted approval for *SEPT9* in the early detection of colon cancer, there is still a lack of FDA-approved methylation biomarkers for breast cancer screening [45]. Gene body methylation plays a pervasive role in the modulation of gene expression and is strongly correlated with both the initiation and progression of malignant tumors [46]. Several gene body methylation biomarkers specific to breast cancer have been previously reported [47,48,49]. However, there is a widespread ongoing debate regarding the precise function of gene body methylation in gene transcription, and the majority of DNA methylation research has focused on CpG islands within gene promoter regions [46]. In the present study, we conducted a genome-wide methylation array analysis to detect hypermethylation at multiple CpG sites in *SRCIN1* in breast cancer patients. Methylation array analysis and quantitative methylation-specific PCR (qMSP) confirmed an increase in the methylation level exceeding 50% at five CpG sites within *SRCIN1* in tissue samples from breast cancer patients in both the TCGA and Taiwanese cohorts. Interestingly, *SRCIN1* hypermethylation did not occur in other cancer types within the Taiwanese and TCGA datasets, confirming the usefulness of *SRCIN1* as a biomarker specifically for breast cancer patients.

Previous studies have revealed diverse diagnostic performances of individual whole-blood DNA methylation markers for early breast cancer detection. S100P exhibited a sensitivity of 71.60% and a specificity of 76.60%. The sensitivity of *BRCA1* detection ranges from 10% to 43%, and the specificity ranges from approximately 85% to 95% [50,51,52]. EPIC methylation array analysis revealed *SRCIN1* hypermethylation in breast cancer patient plasma, and this marker could distinguish between healthy individuals and patients with varying disease stages, with methylation levels increasing with disease severity. *SRCIN1* demonstrated high potential as a noninvasive biomarker with notable sensitivity (78.5%), accuracy (90.3%) and specificity (100%) for early-stage detection. Furthermore, high SRCIN1 methylation also correlated with decreased survival, suggesting its value as a prognostic marker. Further validation in studies with larger cohorts and additional biomarkers could further increase the accuracy of disease detection and prognosis evaluation.

Hypermethylation within the gene body of *SRCIN1* is correlated with elevated gene expression, as evidenced by the high expression of SRCIN1 observed in tumor tissues from both the TCGA and Taiwanese cohorts. Furthermore, we noted a correlation between *SRCIN1* methylation and its expression in our cell line data. Notably, decitabine treatment significantly decreased both the methylation and expression of *SRCIN1*. Previous evidence indicates a correlation between DNA methylation within the gene body and an increased level of gene expression [53,54,55]. High expression of *SRCIN1* was significantly correlated with various clinical variables, including age, histological type, tumor stage, ER status, PR status, HER2 status, TNBC status and luminal A type disease. Analysis of clinical data revealed that increased expression of *SRCIN1* was correlated with early-stage and late-stage breast cancer. Furthermore, its exclusive expression in breast cancer tissue vs. normal breast tissue, suggests that it is a promising candidate therapeutic target [56]. However, for patients with an ERBB2-positive status, the presence of *SRCIN1* is linked to a lower chance of distant metastasis and a clear survival advantage [57]. The role of *SRCIN1* in cancer cell biology is a subject of debate. Preliminary studies have suggested that *SRCIN1* may act either as an oncogene or as a tumor suppressor. *SRCIN1* might suppress β-catenin in the breast cancer stem cell niche, thereby promoting antitumor immune defense [58,59]. Reduced SRCIN1 expression led to decreased cell viability in the MCF-7, T47D and MDA-MB-231 breast cancer cell lines, as confirmed by more than three technical and biological replicates (Figure 5C–E). Furthermore, *SRCIN1* has been implicated in regulating the organization of the actin cytoskeleton in response to cell adhesion and growth factor signals [60]. Considering the latest evidence, the biological impact of *SRCIN1* on cancer, whether it contributes to tumor progression or acts as an oncogene, seems to be influenced by a range of intermediary proteins specific to various cancer types and subtypes. Additionally, to ascertain whether SRCIN1 is a potential drug target, validating its biological function through cell cycle arrest assays and apoptosis analysis is essential, especially in the context of SRCIN1-silenced, untreated and DAC-treated breast cancer cells.

Analysis of our comprehensive dataset, which included three different breast cancer cell lines—MCF7, T47D and MDA-MB231—revealed a significant reduction in cell proliferation upon the silencing of *SRCIN1*. The most significant reduction was observed in MCF7 cells, where estrogen, androgen, progesterone and glucocorticoid receptors are known to play significant roles [61]. To further understand the impact of *SRCIN1* expression on breast cancer, we conducted RNA-seq analysis to explore potential pathways and regulatory proteins linked to *SRCIN1*. These findings were validated using RT‒qPCR. Notably, both RNA-seq analysis and RT‒qPCR revealed a significant decrease in *ESR1*, BCL2 and several cyclin proteins. BCL2 is widely acknowledged as a critical clinical prognostic marker in breast cancer [62]. Cyclins are a family of proteins that regulate cell cycle progression by activating cyclin-dependent kinases [63]. Cyclin A is a prognostic indicator in early-stage breast cancer [64]. Cyclin A2 is an essential regulator of the cell cycle and cell invasion [65]. Additionally, the silencing of *SRCIN1* resulted in the upregulation of proapoptotic proteins and several oncogenes (Appendix A). Consequently, inhibition of SRCIN1 may suppress ESR1 expression, BCL2 expression and pathways involving several cyclin proteins. Aberrant *SRCIN1* methylation in cfDNA from plasma could serve as a companion diagnostic biomarker for SRCIN1 inhibitor therapy in the clinical treatment of breast cancer patients.

## 5. Conclusions

The presence of hypermethylated and circulating methylated *SRCIN1* specifically in breast cancer samples indicates their utility as promising noninvasive biomarkers for the early detection and prediction of disease progression in breast cancer patients. Moreover, hypermethylation of *SRCIN1* is correlated with a poor prognosis, and it has demonstrated high sensitivity and specificity in diagnosis. Attenuation of the hypermethylation of *SRCIN1* to restore its normal expression could be a viable targeted therapeutic strategy.

## Figures and Tables

**Figure 1 biomolecules-14-00571-f001:**
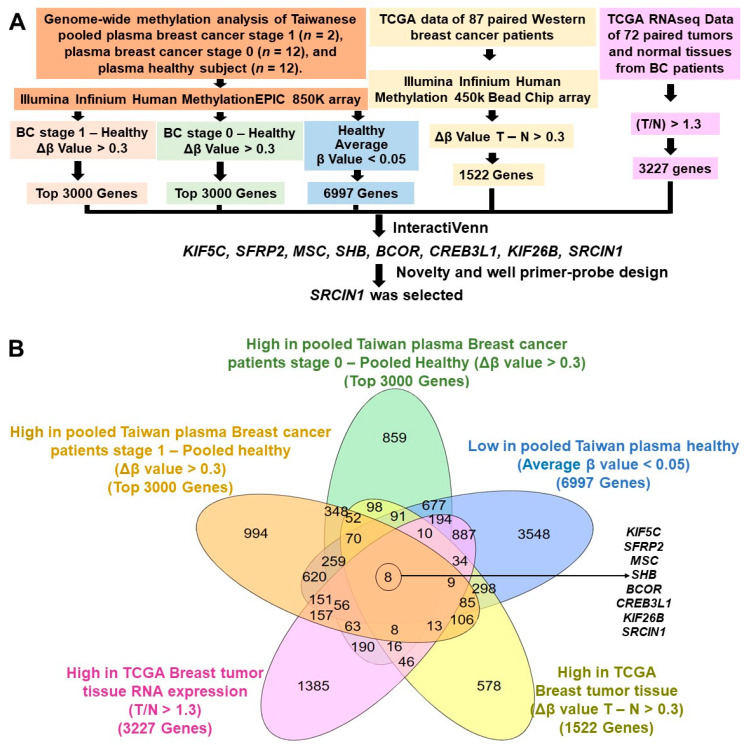
Flowchart of gene selection: (**A**) The criteria and step-by-step flowchart for gene selection. (**B**) Genes identified by screening and their overlap according to InteractiVenn. Δβ, delta beta value. T, tumor tissues. N, normal tissues.

**Figure 2 biomolecules-14-00571-f002:**
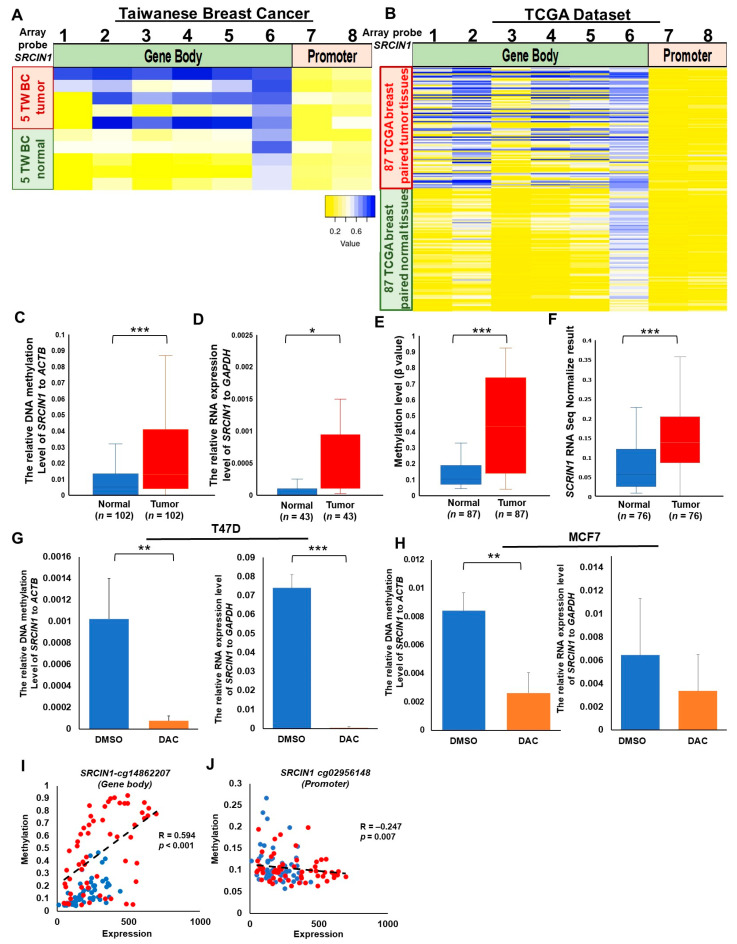
Heatmap of *SRCIN1* methylation patterns and mRNA expression in the Taiwanese and TCGA cohorts and the correlation between methylation and expression levels. (**A**) Heatmap of the *SRCIN1* methylation pattern in 5 breast cancer tissues from the Taiwanese cohort. (**B**) Heatmap of the *SRCIN1* methylation pattern in 87 breast cancer tissues from the TCGA cohort. The highly differentially methylated sites cg14482902, cg01514828, cg03026373, cg14862207 and cg04194674 correspond to array probes 1, 2, 3, 4 and 5, respectively. The genomic positions of the QMSP amplicon are located in exon 1 (+906, +635, −78, −89 and −244) of the *SRCIN1* gene. (**C**) Box plot of *SRCIN1* methylation and (**D**) expression levels in 102 Taiwanese patient tissues. (**E**) Box plot of *SRCIN1* methylation and (**F**) expression levels in 87 TCGA patient tissues. Methylation and mRNA expression levels were analyzed post-treatment with the solvent control DMSO and DAC in T47D breast cancer cells (**G**) and MCF7 breast cancer cells (**H**). The relative DNA methylation levels are shown in the left panel, and the relative mRNA expression levels are shown in the right panel. The data are presented as the mean  ±  SD; * *p*  <  0.05; ** *p*  <  0.01; ****p*  <  0.001. A *t* test was used to assess differences between groups in all experiments; the experiments were performed using at least two biological duplicates and three technical replicates. Pearson correlation analysis of DNA methylation and RNA sequencing data for 87 paired adjacent normal and cancer tissue samples in gene bodies region (**I**) and gene promoter regions (**J**). Red dots indicate tumor tissue, while blue dots signify normal tissue.

**Figure 3 biomolecules-14-00571-f003:**
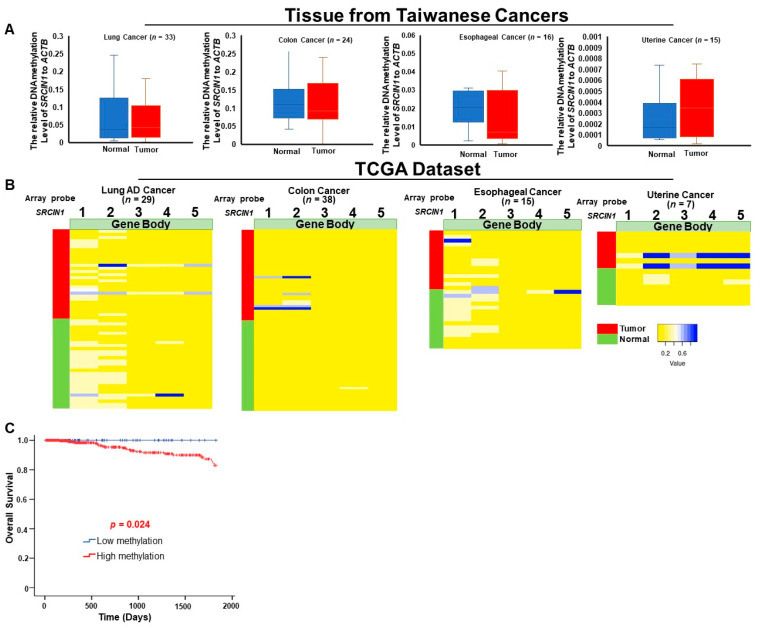
*SRCIN1* gene alterations specific to breast cancer correlate with poorer survival outcomes: (A) Differentially methylated CpG heatmap of *SRCIN1* in paired lung cancer, colon cancer, esophageal cancer and uterine cancer from Taiwan; (**B**) in the TCGA dataset, the CpG sites in gene body regions, 1 = cg14482902, 2 = cg01514828, 3 = cg03026373, 4 = cg14862207, 5 = cg04194674; (**C**) Kaplan–Meier survival curves were constructed to compare overall survival between breast cancer patients with high and low methylation of *SRCIN1.* The data are presented as the mean  ±  SD. A *t* test was used to assess differences between groups in all experiments. The experiments were performed using three technical replicates.

**Figure 4 biomolecules-14-00571-f004:**
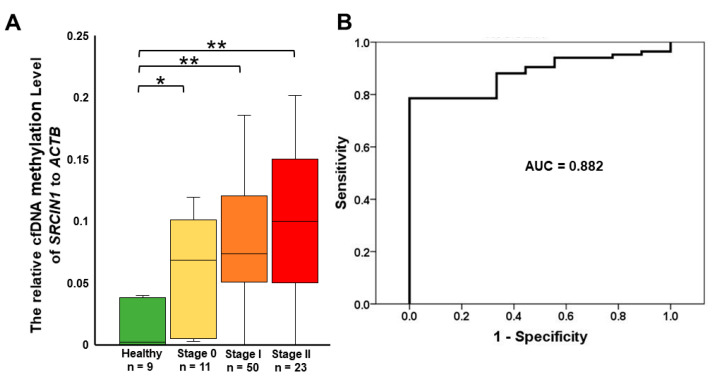
ROC curve analysis and detection of *SRCIN1* cfDNA methylation in plasma samples for the early diagnosis of breast cancer in patients from Taiwan and the USA: (**A**) Relative cfDNA methylation level of *SRCIN1* after normalization to the *ACTB* gene; (**B**) ROC curve for the prediction of breast cancer. The aberrant methylation of *SRCIN1* in cfDNA was quantitatively assessed via qMSP and normalized to that of *ACTB*, with ratios exceeding 0.04 indicating positive *SRCIN1* methylation. The data are presented as the mean  ±  SD; * *p*  <  0.05; ** *p*  <  0.01. A *t* test was used to assess differences between groups in all experiments. The experiments were performed using three technical replicates.

**Figure 5 biomolecules-14-00571-f005:**
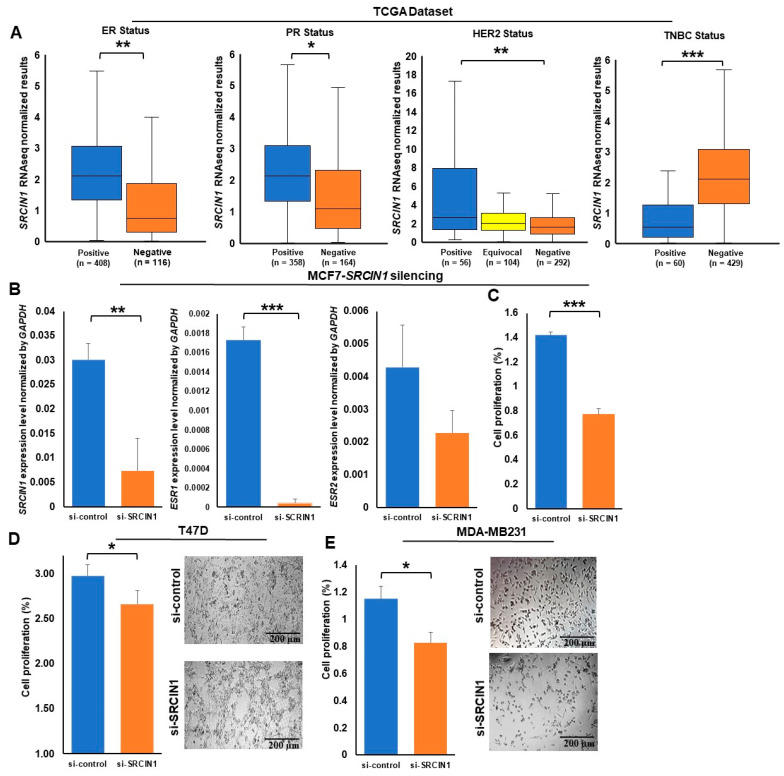
*SRCIN1* expression in breast cancer cell proliferation and its correlation with hormonal receptor and HER2 status: (**A**) box plot of *SRCIN1* expression levels correlated with ER, PR, HER2 and TNBC status data generated from the TCGA dataset; (**B**) *ESR1* and *ESR2* expression after knockdown of *SRCIN1* in the MCF7 cell line; (**C**) the relative proliferation of MCF7 cells; (**D**) images of T47D breast cancer cell lines after transfection with si-*SRCIN1* and the relative proliferation of T47D cell lines; (**E**) images of MDA-MB231 breast cancer cell lines after transfection with si-*SRCIN1* and the relative proliferation of MDA-MB231 cell lines; *GAPDH* was used as an internal control. The data are presented as the mean  ±  SD; * *p*  <  0.05; ** *p*  <  0.01; *** *p*  <  0.001. A *t* test was used to assess differences between groups in all experiments; the experiments were performed using at least two biological duplicates and three technical replicates.

**Figure 6 biomolecules-14-00571-f006:**
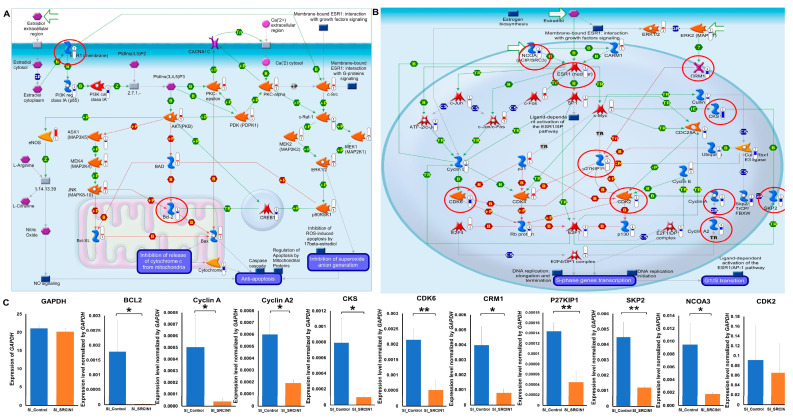
Network analysis of *SRCIN1* involvement in estrogen receptor downstream signaling pathways and the cyclin protein family: (**A**) MetaCore analysis of estrogen receptor downstream signaling pathways and apoptosis; (**B**) RNA-seq; (**C**) RT‒PCR after transfection with si-*SRCIN1* in MCF7 cells indicating decreases in Cyclin A, Cyclin A2, CKS, CDK6, CRM1, P27KIP1 SKP2, NCOA3 and CDK2. The data are presented as the mean  ±  SD; * *p*  <  0.05; ** *p*  <  0.01. A *t* test was used to assess differences between groups in all experiments.

**Table 1 biomolecules-14-00571-t001:** *SRCIN1* methylation levels and mRNA expression in relation to the clinical parameters of Taiwanese patients with breast cancer ^a^.

Characteristics		Total ^b^ (%)	Methylation ^c^		*p* Value	Total (%)	Expression ^d^			*p* Value
			Low (%)	High (%)			Low (%)	Normal (%)	High (%)	
**Overall**		102	39 (38.2)	63 (61.7)		43	4 (9.3)	18 (41.8)	21 (48.8)	
**Stage**		96	36 (37.5)	60 (62.5)	0.427	41	4 (9.8)	16 (39.0)	21 (51.2)	0.227
	I and II	69 (71.9)	25 (36.2)	44 (63.8)		33 (80.5)	3 (9.1)	15 (45.5)	15 (45.5)	
	III and IV	27 (28.1)	11 (40.7)	16 (59.3)		8 (19.5)	1 (12.5)	1 (12.5)	6 (75.0)	
**Histological Type**		98	37 (37.8)	61 (62.2)	0.385	42	4 (9.5)	17 (40.5)	21 (50.0)	0.471
	IDC	96 (98)	37 (38.5)	59 (61.5)		41 (97.6)	4 (9.8)	16 (39.0)	21 (51.2)	
	ILC	2 (2)	0 (0)	2 (100)		1 (2.4)	0 (0.0)	1 (100)	0 (0)	
**Tumor size**		97	36 (37.1)	62 (62.9)	0.089	41	4 (9.5)	17 (40.5)	20 (50.0)	0.830
	T0–T1	25 (25.8)	6 (24.0)	19 (76.0)		16 (39.0)	2 (12.5)	7 (43.8)	7 (43.8)	
	T2–T4	72 (74.2)	30 (41.7)	42 (58.3)		25 (61.0)	2 (8.0)	10 (40.0)	13 (61.0)	
**Lymph node involvement**		94	34 (36.2)	60 (63.8)	0.507	40	4 (10.0)	17 (42.7)	19 (47.5)	0.567
	N0	40 (42.6)	14 (41.2)	26 (65.0)		22 (55.0)	2 (9.1)	11 (50.0)	9 (40.9)	
	N1–N3	54 (57.4)	20 (37.0)	34 (63.0)		18 (45.0)	2 (11.1)	6 (33.3)	10 (55.6)	
**Subtypes**					0.352					0.935
	Luminal A	15 (17.4)	8 (53.3)	7 (46.7)		5 (17.2)	0 (0.0)	2 (40.0)	3 (60.0)	
	Luminal B	37 (43.0)	13 (35.1)	24 (64.9)		11 (37.9)	1 (9.1)	4 (36.4)	6 (54.5)	
	HER2	18 (20.9)	5 (27.8)	13 (72.2)		7 (24.1)	0 (0.0)	3 (42.9)	4 (57.1)	
	TNBC	16 (18.6)	8 (50.0)	8 (50.0)		6 (20.1)	0 (0.0)	2 (33.3)	4 (66.7)	
**Tumor markers**										
**Estrogen**		93	37 (39.8)	56 (60.2)	0.356	38	2 (5.3)	16 (42.1)	20 (52.6)	0.497
	Positive	62 (66.7)	26 (41.9)	36 (58.1)		25 (65.8)	2 (8.0)	11 (44.0)	12 (48.0)	
	Negative	31 (33.3)	11 (35.5)	20 (64.5)		13 (34.2)	0 (0.0)	5 (38.5)	8 (61.5)	
Progesterone		93	37 (39.8)	56 (60.2)	0.465	38	2 (5.3)	16 (42.1)	20 (52.6)	0.762
	Positive	51 (54.8)	21 (41.5)	30 (58.8)		24 (63.2)	1 (4.2)	10 (41.7)	13 (54.2)	
	Negative	42 (45.2)	16 (43.2)	26 (61.9)		14 (36.8)	1 (7.1)	6 (42.9)	7 (50.0)	
HER2		92	36 (39.1)	56 (60.9)	0.094	37	1 (2.7)	16 (43.2)	20 (54.1)	0.542
	Positive	55 (59.8)	18 (32.7)	37 (67.3)		18 (48.6)	1 (5.6)	7 (38.9)	10 (55.6)	
	Negative	37 (40.2)	18 (48.6)	19 (51.4)		19 (51.4)	0 (0.0)	9 (47.4)	10 (52.6)	
Ki-67		84	33 (39.3)	51 (60.7)	0.441	30	2 (6.7)	15 (50.0)	13 (43.3)	0.725
	>14%	59 (70.2)	24 (40.7)	35 (59.3)		22 (73.3)	1 (4.5)	11 (50.0)	10 (45.5)	
	<14%	25 (29.8)	9 (36.0)	116 (64.0)		8 (26.7)	1 (12.5)	4 (50.0)	3 (26.7)	

^a^ These results were analyzed via Pearson’s X2 test. ^b^ For some categories, the number of samples (n) was lower than the overall number analyzed because clinical data were unavailable for those samples. ^c ^*SRCIN1* methylation was considered to indicate hypermethylation when the *SRCIN1* methylation level was greater than two in breast tumors compared to adjacent normal breast tissues. ^d^
*SRCIN1* expression was considered high when the *SRCIN1* expression level was greater than two in breast tumors compared to adjacent normal breast tissues.

**Table 2 biomolecules-14-00571-t002:** The sensitivity of *SRCIN1* cfDNA methylation for the early prediction of breast cancer in patients from Taiwan and the USA ^a^.

		*SRCIN1* Methylation	
	Total N	Low N (%)	High ^b^ N (%)
	93	27 (29.0)	66 (71.0)
Healthy	9	9 (100.0)	0 (0.0)
Stage 0	11	4 (36.4)	7 (63.6)
Stage I	50	10 (20.0)	40 (80.0)
Stage II	23	4 (17.4)	19 (82.6)

^a^ The results were tested in 9 plasma samples from healthy controls and 84 plasma samples from breast cancer patients. ^b^ When the methylation level ratio of *SRCIN1* relative to that of the *ACTB* gene was greater than 0.04, the sample was considered to have high methylation.

## Data Availability

The data generated in this study are available from the corresponding author upon reasonable request.

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
