# Peer review of "Hypermethylation of the Gene Body in SRCIN1 Is Involved in Breast Cancer Cell Proliferation and Is a Potential Blood-Based Biomarker for Early Detection and a Poor Prognosis"

_biomolecules, 2024, doi:10.3390/biom14050571_

Round 1
Reviewer 1 Report
Comments and Suggestions for Authors
Dear Author,
Thanks for submitting your manuscript. In the study, you proposed a strategy to screen for epigenetic markers for breast cancer and identified the hypermethylation status in the gene body of SRCIN1 in patient samples. Furthermore, you assessed the potential of SRCIN1 as a biomarker for breast cancer diagnosis and prognosis by investigating its methylation levels, patterns, and biological functions. Overall, the work is well-designed and effectively presented. The methods employed were appropriate, and the results strongly support the conclusions drawn. I have outlined several suggestions for improvement below:
Minors:
- Correct the typo in line 119 where "20.0" should be "2.0".
- Use the full name of ACTB for its first occurrence in line 145.
- Rewrite line 365 for clarity.
- Italicize "SRCIN1" in lines 339 and 435.
- Provide justification for the choice of a cutoff of 0.04 in line 152.
- In line 289, clarify if 1.3 in log-scale and justify the choice of this value.
- Consider using R instead of Excel for creating heatmaps.
- Include legends for Figure 2I/2J.
Majors:
- According to Figure 1B, the SRCIN1 gene was identified by selecting overlapping genes from hypermethylated and up-regulated genes (RNA-seq). Why was the analysis limited to up-regulated genes only?
- Line 152 mentioned a ratio of 0.04 used to select positive methylation. However, in the DMSO groups (Figure 2H/G) for the two breast cancer cell lines, the relative methylation level is far below 0.04. Please comment on this.
Best,
Comments on the Quality of English LanguageEnglish is good.
Author Response
Response to Reviewer comments:
Reviewer 1:
Minor Reviewer comments:
- Correct the typo in line 119 where "20.0" should be "2.0"
Response: We apologize for the typo on line 119 and have corrected "20.0" to "2.0" as suggested. Thank you for pointing this out.
- Use the full name of ACTB for its first occurrence in line 145.
Response: The full name of actin beta (ACTB) has been included at its first mention in line 146. Thank you for this correction. - Rewrite line 365 for clarity.
Response: The sentence on lines 368 to 370 has been rephrased for clarity: "Methylation and mRNA expression levels were analyzed post-treatment with the solvent control DMSO and DAC in T47D breast cancer cells (G) and MCF7 breast cancer cells (H)." Thank you for your input. - Italicize "SRCIN1" in lines 339 and 435.
Response: The gene name "SRCIN1" has been italicized in lines 342 and 448 as requested. Thank you for highlighting this. - Provide justification for the choice of a cutoff of 0.04 in line 152.
Response: We've provided a rationale for selecting 0.04 as the cutoff in line 417 to 422. This cutoff is based on the observation that the methylation level of SRCIN1 in cfDNA, normalized with ACTB, is below 0.04 in healthy subjects (as shown in Figure 4A). We've included a detailed explanation from lines 417 to 422 to clarify that establishing a biomarker with high specificity for breast cancer detection, which has a lower cfDNA methylation background in healthy individuals, is of primary importance.
The following description has been added to the manuscript to provide justification for the selection of the 0.04 cutoff:
“To ensure the high specificity of the blood biomarker for detecting breast cancer, it is critical to have a low cfDNA methylation background in healthy subjects. Our investigations revealed that the methylation level of SRCIN1 in cfDNA from plasma, when normalized with ACTB used as the cfDNA control, is consistently below 0.04 across all samples from healthy individuals (Figure 4A). Consequently, we have identified 0.04 as the threshold to differentiate between positive and negative tests.”
- In line 289, clarify if 1.3 in log-scale and justify the choice of this value.
Response: We have added clarification in line 289 that the “1.3 fold change” value was chosen as the minimal threshold to identify genes with weak overexpression in RNA levels compared to normal breast tissues, which may serve as potential drug targets. A detailed justification, including references for the selection of the 1.3 fold change threshold, has been incorporated into line 287 to 290 of the manuscript: To identify potential oncogenes for drug targeting, we have selected a minimal threshold of a 1.3-fold change to distinguish genes that exhibit overexpression at the RNA level compared to normal breast tissue expression profiles.
- Consider using R instead of Excel for creating heatmaps.
Response: We have addressed the suggestion by utilizing the heatmapper program (http://www.heatmapper.ca/expression/) to generate the heatmaps. The heatmaps have been customized to represent high, middle, and low DNA methylation levels using a gradient from blue to yellow, with high DNA methylation levels indicated in black. The updated heatmaps have replaced the previous versions in Figure 2A, 2B, and Figure 3B of the manuscript. We've included a detailed explanation in the materials and methods sections from lines 240 to 243.
- Include legends for Figure 2I/2J.
Response: We have included the legends for the Figure 2I/2J in lines 374-379. Thank you for highlighting this.
Major Reviewer comments:
- According to Figure 1B, the SRCIN1 gene was identified by selecting overlapping genes from hypermethylated and up-regulated genes (RNA-seq). Why was the analysis limited to up-regulated genes only?
Response: As described in line 81 of the manuscript, the aim of this study is to identify plasma-based biomarkers for the early diagnosis and prognosis of breast cancer and to pinpoint potential drug targets. Strategically selecting genes that are both hypermethylated and up-regulated enables the identification of overexpressed oncogenes that can be detected through methylated cfDNA. This could potentially lead to viable options for companion diagnostics and associated drug targets, thereby fulfilling one of the primary objectives of our research. We also articulate our rationale in line 267 to 269: To identify novel potential biomarkers for early prediction and drug targets in breast cancer patients from both Taiwanese and Western cohorts, we have established three criteria to screen for targets.
- Line 152 mentioned a ratio of 0.04 used to select positive methylation. However, in the DMSO groups (Figure 2H/G) for the two breast cancer cell lines, the relative methylation level is far below 0.04. Please comment on this.
Response: The 0.04 cutoff for the selection of positive or negative results was established based on the methylation level of SRCIN1 in cfDNA from plasma, primarily derived from tumor cells undergoing apoptosis, necrosis, or secretion, and then normalized with ACTB. ACTB serves as the cfDNA control and reflects the methylation background from a variety of body tissues and fluids; this level is consistently below 0.04 in all samples from healthy individuals. In contrast, the methylation levels observed in the DMSO groups (Figure 2H/G) of the T47D and MCF7 breast cancer cell lines differ because the ACTB normalization in this context comes from the genomic DNA of the cell lines themselves, not cfDNA. Consequently, the methylation levels of SRCIN1 and the quantity of ACTB in these cell lines differ significantly from those in cfDNA from plasma, leading to the observed discrepancy.
Reviewer 2 Report
Comments and Suggestions for Authors
Shen and colleagues presented an interesting study on the involvement of epigenetic in breast cancer. The authors analyzed tissue and plasma of patients from two different hospitals and the plasma samples from another country. The data collected allowed to identify SRCIN1 as a potentially biomarker, related to methylation, for breast cancer. The results were integrated and validated analyzing the methylation status of SRCIN1 in TCGA data. The authors tried to explain the biological function of SRCIN1 correlating its hypermethylation, and consequently up-regulation, with estrogen receptor ESR1 and BCL2 genes and some members of cyclin protein family. In this way, they supposed the involvement of SRCIN1 hypermethylation in viability of cancer cells. In my opinion MTT is a good method to assess viability of cells, but not enough to determine the involvement of SRCIN1 in cell proliferation or apoptosis. Thus, I suggest the authors to complete the biological validation of SRCIN1 role in cell survival performing cell cycle arrest and/or apoptosis analysis on SRCIN1-silenced breast cancer cells and after treatment with DAC. I think that these additional experiments will increase the value and the comprehension of the role of SRCIN1 methylation in breast cancer.
I need to highlight some minor revision to authors. Firstly, I did not find the amount of DNA and cfDNA subjected to bisulfite conversion, so I suggest specifying the input of the reaction. Secondly, the authors should indicate the concentration of DCA used for cell treatments to give the readers a more complete chart of experiments. Additionally, I suggest the authors to replace the first reference with a more recent one about the global burden of breast cancer (i.e. Current and future burden of breast cancer: Global statistics for 2020 and 2040, Breast 2022, 66, 15-23).
Nevertheless, beyond all the above comments, I think the manuscript may be evaluated for publication on Biomolecules journal after some minor revisions.
Author Response
Response to Reviewer comments:
Reviewer 2:
Point 1: The Reviewer suggests the authors to complete the biological validation of SRCIN1 role in cell survival performing cell cycle arrest and/or apoptosis analysis on SRCIN1-silenced breast cancer cells and after treatment with DAC.
Response: We concur with the reviewer on the necessity of fully validating SRCIN1's role in cell survival. Accordingly, we will undertake additional experiments to evaluate the impact of SRCIN1 silencing on cell cycle arrest and apoptosis in breast cancer cells, both before and after DAC treatment. These studies will not only bolster the evidence for SRCIN1's role in these essential cellular processes but also support its candidacy as a drug target. We have elaborated on this planned research in the discussion section, specifically at line 543 to 546 of our manuscript, stating: To ascertain whether SRCIN1 is a potential drug target, validating its biological function through cell cycle arrest assays and apoptosis analysis is essential, especially in the context of SRCIN1-silenced, untreated, and DAC-treated breast cancer cells.
We appreciate this insightful suggestion, which will certainly enrich the depth and rigor of our research findings.
Point 2: The reviewer suggests that the manuscript should specify the quantity of DNA and cfDNA used for bisulfite conversion and also requests that the concentration of DAC applied in cell treatments be detailed to provide a comprehensive overview of the experimental procedures.
Response: We apologize for the oversight and have now specified the quantities of DNA and cfDNA for bisulfite conversion and the concentration of DAC for cell treatments in the manuscript. We have added the description in line 135 for quantity of DNA and cfDNA and line 196 for DAC concentration. Thank you for pointing out the need for this detail.
Point 3: The reviewer suggests that the authors to replace the first reference with a more recent one about the global burden of breast cancer (i.e. Current and future burden of breast cancer: Global statistics for 2020 and 2040, Breast 2022, 66, 15-23).
Response: We thank the reviewer for the recommendation to update our references with more current data on the global burden of breast cancer. We have replaced the initial reference with the one suggested, "Current and future burden of breast cancer: Global statistics for 2020 and 2040" from Breast 2022, volume 66, pages 15-23, to reflect the most recent statistics and projections.